# Clustering knowledge and dispersing abilities enhances collective problem solving in a network

Charles J. Gomez [1]* & David M.J. Lazer[2,3]

Diversity tends to generate more and better ideas in social settings, ranging in scale from small-deliberative groups to tech-clusters and cities. Implicit in this research is that there are knowledge-generating benefits from diversity that comes from mixing different individuals, ideas, and perspectives. Here, we utilize agent-based modeling to examine the emergent outcomes resulting from the manipulation of how diversity is distributed and how knowledge is generated within communicative social structures. In the context of problem solving, we focus on cognitive diversity and its two forms: ability and knowledge. For diversity of ability, we find that local diversity (intermixing of different agents) performs best at all time scales. However, for diversity of knowledge, we find that local homogeneity performs best in the long-run, because it maintains global diversity, and thus the knowledge-generating ability of the group, for a longer period.

---

[1] The City University of New York—Queens College, Flushing, USA. [2] Northeastern University, Boston, Massachusetts, USA. [3] Institute for Quantitative Social Science, Harvard University, Cambridge, Massachusetts, USA. *email: charles.gomez@qc.cuny.edu

For a given amount of diversity in a social system, is it better for similar types to be grouped together? There is a common thread across multiple literatures that there are generative benefits that come from the mixing of different types of individuals and their ideas, allowing for the identification of novel solutions[1–6]. Diversity tends to generate more and better ideas in social settings[7–13], ranging from small-deliberative groups[13–17] to tech clusters and cities that are viewed as more creative and more efficient[13,18,19]. Implicit in this research is that *mixing* different individuals, ideas, and perspectives helps to identify new knowledge[1–6]. These benefits largely stem from *cognitive* diversity[9], which reflects the various and distinctive ways we think about and interpret the world to solve problems.

Our focus is how the distribution of cognitive diversity (henceforth diversity) impacts the performance of problem-solving collectives, such as small-deliberative groups, knowledge-intensive industries, and even diverse, modern democracies[15,20–26]. Indeed, this is a growing issue of concern with far-reaching implications. For instance, innovation in knowledge-intensive organizations is often heralded as the result of the synergies across diverse, multidisciplinary efforts, like engineers and product designers working on the latest handheld mobile device. Here, the distribution of diversity tends toward local heterogeneity with extensive intermixing, which is shown to generate innovative ideas and enhance adaptability and error detection[6,12,13,15,16,27–29].

While most of the preceding literature on the role of diversity within groups has been substantially explored in the social sciences, computer scientists, geneticists, and operations researchers have explored diversity in learning and search algorithms[8–10,30–32]. Findings from both empirical research and agent-based models consistently suggest that diversity supports collective creativity in problem solving. However, these efforts focus on the impact of having a fixed amount of global (or group-level) diversity in a system and overlooks the effects of the distribution of that diversity within a system[7,8,10,33,34]. To the best of our knowledge, no work to date has tested how instead the distribution of a fixed amount diversity shapes overall outcomes. As such, our focus takes group-level diversity as a constant and varies the mixing across types within groups. We build upon the rich tradition of agent-based modeling[35,36] and focus on the impact of how diversity is distributed within communicative social structures[20,33,37–39]. Following Lazer and Friedman[39] and Hong and Page[7], we utilize agent-based models situated in social networks to examine the emergent consequences of agent interaction for parallel problem solving. Agent-based models are particularly useful and appropriate to explore this because, as "computational experiments,"[36] they (1) yield empirically testable propositions across a wide range of settings (e.g., online and small-group experiments, etc.); and (2) allow us to quickly and efficiently test for causality across various permutations of structure (e.g., networks) and agency (e.g., agents with different heuristics)[36,39]. They can also inductively explore how incremental changes in agent behavior have non-linear and synergetic system-wide results[36]. Furthermore, social networks are useful as their structures can prohibit or enhance access to novel information[40–42].

That is, what impact do structures have on how connected are actors of different types to each other? At a trivial level, a "group" in which different types of individuals *never* communicated with each other presumably does not get the creative dividends from its apparent diversity. However, beyond "there should at least be a single tie connecting different types of actors," the literature offers little guidance as to the impact of connectivity between different types of actors in a collective with a given level of diversity.

To that end, our analytic question is simple: How does the distribution of types of agents within a network affect the collective performance of the system? Is it better to have similar types intermixed together or relatively separated? Our approach to modeling the distribution of diversity, or what we call the rate of *intermixing*, varies the extent to which identical agents or identical ideas are clustered together. Our mode of inference is experimental[36]—we have a number of starting conditions where we manipulate whether the networks are intermixed or not, generate an arbitrarily large number of cases, and compare outcomes.

From a modeling perspective, this yields two subsidiary questions: (1) how to model diversity and (2) how to model its intermixing within a network? We focus on two types of diversity: *ability* and *knowledge*. Our focus here is on problem solving and how distribution of knowledge and ability within the network of a group contributes to the performance of the group. In the context of problem solving, we define *knowledge* as knowing that a particular solution produces a given level of performance, and *ability* as the capacity to act upon given knowledge to produce new knowledge. For instance, consider the smallest of possible social systems, made up of two agents. They might have the same set of knowledge at time $t$ (i.e., low diversity of knowledge), but different capacities to incrementally move toward new knowledge (i.e., high diversity of ability). Alternatively, the two agents might have different states of knowledge (i.e., high diversity of knowledge), but share the same capacity to move toward new states of knowledge (i.e., low diversity of ability). Our question here is simply: is it best for agents with similar abilities (knowledge) to be placed near or far from each other in the group's network?

The underlying metaphor of the model for this paper is that an essential truth about the world is that there are sets of actors with diverse knowledge and skills that are all trying to solve the same or similar complex problem. That problem might be high schools trying to educate their students; academics trying to build successful careers; or states trying to produce public policy that improves the health of their citizens. In each of these cases, performance of an actor is not substantially affected by the performance of other actors (e.g., success with the current opioid crisis in one state probably does not have a great direct effect on the opioid crisis in other states), there is some feedback from the environment, and there is the potential to learn from other actors. This is the general framework of "parallel problem solving."[39] Prior research, for example, highlights that the efficiency of communication within a system of parallel problem solvers increases the initial speed of improvement of solutions within the system, but at a long-run cost of performance due to reduced exploration for new solutions. Working from this motivating metaphor, we offer formalizations of complex problems, social networks through which actors learn from each other, and diversity.

A complex problem is one that has many dead ends where an agent can get stuck (local optima), and a few "excellent" solutions. It is, according to Herbert Simon[43] (p. 198), "the capacity of the human mind for formulating and solving complex problems is very small compared with the size of the problems whose solution is required for objectively rational behavior in the real world—or even for a reasonable approximation to such objective rationality." Complex problems can be visualized as a rugged terrain, like a mountain range, and agents can be thought of as myopic climbers on these terrains seeking a global optimum: the highest possible peak in the terrain, and many lower peaks that can trap climbers at a lower altitude. The key attribute of any problem space is how rugged it is, which is an indication of how difficult it is for agents to find the optimal solution. This determines how rugged the

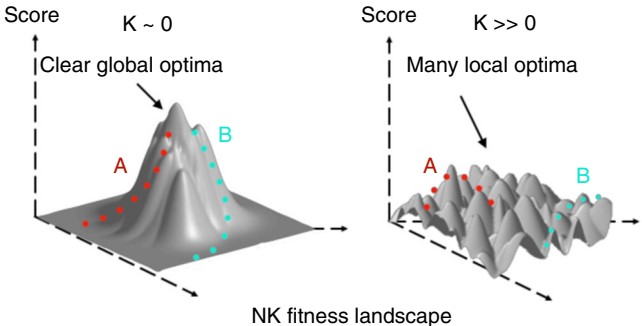

**Fig. 1** NK space. Hypothetical plot of two NK problem spaces with varying levels of *K*

space is: essentially, how many (lower) peaks there are that climbers might get stuck on.

The NK problem space has been a standard formalization used to model complex problems[44–50] (We explain many of the design and analytic decisions in the "Methods" section, but offer a conceptual overview here.) An NK space is an *N*-dimensional space, which are each associated with one of several potential scores. What determines the exact score for any particular point in this space depends on a fixed tuning parameter *K* that sets up the NK space as either more rugged (higher values of *K*) or smoother (lower values of *K*). In an NK space with a low value of *K*, agents with different search strategies (i.e., say some search strategy A and B) can readily find the best solution (global optimum) with ease. However, in an NK space with a high value of *K*, agents meander about local optima that are less profitable in terms of score than the easily achievable low-*K* global optimum space. So much so, agents with narrow search capabilities can get stuck in local valleys. Figure 1 offers a stylized characterization of a comparison of two NK spaces, one with a low *K*, and one with a high *K*.

Every point in an NK space is defined by an address, represented by a sequence of length *N* of 1s and 0s. Agents in a simulation are given this address and manipulate it in various ways iteratively and myopically (e.g., changing one of the random 1s in the address to a 0 or vice-versa, for instance) to jump to a new point in the space that might have higher score. NK spaces that are very simple allow agents to quickly jump to points with higher scores, and those that are very complex are typically very difficult for agents for find paths that lead to the global optimum. Hence, tuning *K* allows us to create a challenging but searchable space to explore. The scores for each NK space are normalized such that the global optimum is assigned a score of 1 and the worst possible solution is assigned a score of 0.

NK spaces are analytically useful for agent-based models but are conceptually abstract and difficult to ground to real-world examples. Here, we can instead understand an NK problem's complexity in terms of how incremental contributions made by any subset of activities are also contingent on other activities. Thus, a solution that an agent comes up within the NK space—captured by a unique sequence of 1s and 0s—reflects a basket of activities people undertake to solve a real-world problem, where a 1 represents the presence of an activity and a 0 its absence.

There are many examples of social collectives solving a common problem that are analogous to the NK space setup. In these instances, the solutions and performances are visible at least some subset of other actors, and there are no positive or negative externalities from one actor to another (beyond the informational spillover). For instance, consider national ministries of health of various countries. They all have different approaches to healthcare (e.g., insurance markets like in Germany or Austria or a single-public option like in Canada or the U.K.). And the improvement or decline of one country's healthcare does not adversely affect or hamper the healthcare of another. The interest of some countries in the policies employed by other countries may be limited by physical adjacency or similar political systems or cultures, so their information about the "problem space" is more localized.

To expound on this a bit further with respect to the NK space, consider superintendents of high school districts[50]. Here, superintendents can learn from the best practices and performance of superintendents from other high school districts, specifically where they can see the practices and performance of other local schools. The improvement or decline of one school or district does not necessarily affect another. Furthermore, superintendents are diverse in that they employ approaches that differ from other superintendents as to how they view the schools in their districts: Some may view their schools more as administrators, while others view it more pedagogically. As such, one "activity" that a superintendent as an administrator might employ at their schools is to hire better teachers or to change to block scheduling. Alternatively, a pedagogically oriented superintendent may work to improve the curriculum collaboratively with local teachers. However, these activities do not exist independently. Instead, they have synergies among them such that this approach only works in the presence of other tactics (e.g., certain curriculum improvements only work with block scheduling.). As such, the NK space captures the interaction of activities and these synergies. In other words, it is contingent on "sub-problems" related to the wider goal. Nevertheless, pulling on the diverse skillset and solutions across the department, finding the optimized solution is possible.

The severity of this problem's complexity is not additive or linear in nature, but *interdependent* on all of the sub-problems that need to be solved for this to work. Furthermore, adopting a solution might have unintended, if not non-linear, consequences to the system's overall outcome that were not obvious or foreseeable. For instance, allowing students to use their mobile phones in schools might seem like a good idea to improve class engagement through other media, but an unintended consequence might be distraction or discouraging deeper critical thinking. The objective of the NK space is to capture the complex interaction among activities that yield performance. The NK problem space's popularity in modeling human decision-making stems from its verisimilitude with the complex and multidimensional problems that face problem-solving tasks, and because researchers can easily generate a large number of statistically similar problem spaces for robustness checks[7,36,51]. (We note, however, in order to make sure our results are not an artifact of the idiosyncrasies of the NK problem space, we replicated all of our results in another rugged problem space, the Traveling Salesperson Problem—findings available in Supplemental Methods in Supplementary Fig. 2 through 9 and 11).

Typically, in any search problem we assume that agents can only view a subset of the landscape (i.e., there are no omniscient agents that can simply view the entire landscape and pick the global optimum). Instead, we assume that people generally are very myopic with respect to the choices they make (i.e., they can see the consequences of only the neighborhood of choices around their status quo choice, and not the longer-term possibilities that those choices open up or foreclose). In our model, we therefore endow our agents with search capacities in the problem space that allows view of only a tiny fraction of the entire problem space.

To that end, we build on the paradigm of "parallel problem solving," where agents seek to improve their solution to a complex problem they confront[39] and the performance of one agent does not adversely affect another. For instance, principals whose

expertise is in school administration and principals whose expertise is in pedagogy all work in parallel and collaboratively by sharing best practices to improve their high schools. For example, the pedagogically oriented principals find that flipped-classrooms improve standardized test performance, which the administratively-oriented principals adopt in their schools and then improve upon in their own way. (The alternative case where this leads to worse outcomes also shows how one good solution discovered by one agent may in fact lead to a worse solution by a different agent with different search abilities; and globally leads to a worse overall solution as compared to the case where peer-review was effective.)

Our motivation behind developing our model in this way is that it ought to be authentic, robust, and replicable, where simplicity is a critical means to all of those ends[36]. On the one hand, much of what happens in the real world is glossed over with this level of abstraction: For instance, the value of diversity might come at the cost of mistranslations (e.g., administrative principals do not readily speak the same language as pedagogical principals). On the other hand, a model that directly accounted for all of the complex dynamics of interpersonal interactions associated with large-scale projects would have low external validity to other similar situations, like cross-functional brain-storming sessions in an innovative tech startup. Here, abstracting the model may not be a catch-all for all of the details and richness found in real-world settings, but allows us to manifest the effects of diversity distributions with limiting contextualization to any specific case or situation for future empirical work to expand upon.

**The network and experimental setup**. For our experiment, we distribute one hundred agents in a communicative torus network[52–54], where every agent in the torus network has four neighbors: two agents to their immediate left and right. We do so because the torus network is a useful communication model that strongly differentiates between local and systemic proximity[55]. In other words, agents are limited to a "local" neighborhood of other agents that can be highly diverse or homogeneous, but can be readily exposed to a distant part of the "global" network by randomly "rewiring" a connection to another agent far away. (The latter is part of our robustness checks in the Supplemental Methods and in the Supplementary Figs. 5, 6, 10, and 11.) This allows us to better isolate the effects of diversity in this setup.

Each agent begins with a random piece of knowledge of a very large problem space they will explore (e.g., the address to some random starting point in the NK space), with the goal of finding the best possible solution (a point with a higher score). These pieces of knowledge (referred to as their "state of knowledge") are randomly distributed such that every agent receives a unique one for each problem space they will explore. (As explained in more detail later and in the Supplementary Methods, while they are randomly and uniquely assigned to agents in the network for each problem space, they are the same across experimental setups to render them comparable.) Our focus here is how various distributions of diversity affect parallel problem-solving performance. However, if everyone in the department constantly communicated with everyone else, then distributing people with similar-types or cross-functionally is a moot point, as everyone can communicate with everyone else.

Starting in round 1, every agent looks at the current states of knowledge (i.e., solutions and associated performance) of its neighbors (i.e., the other agents it is connected to in the network). As inspired by March[20], agents then engage in one of two behaviors: *exploitation* and, if that fails to produce a better result, *exploration*. First, by *exploitation*, the agent will copy the current state of knowledge of a more successful neighbor (i.e., exploits the

neighbor's state of knowledge at time *t*), if one exists. If none of the agent's neighbors' states of knowledge are currently better at time *t*, the agent will attempt to *explore* for better states of knowledge. The agent does so by incrementally changing its *current* state of knowledge. This is how we define our first form of diversity, ability, which we discuss and set up in the next section. If its new (read: *experimental*) state of knowledge that results from a random incremental change has a higher score than its *current* state of knowledge, the agent adopts this *new* state of knowledge as its own. Otherwise, the agent remains with its current solution.

All agents in the network engage in this behavior in a single round. The next round of the simulation ($t + 1$) repeats the aforementioned steps, but now agents begin with their *current* state of knowledge from the previous round *t*, where they found a new solution from exploring the problem space, copied a neighbor with a better solution, or continued with their status quo solution. This process continues for a number of rounds until the entire system reaches an equilibrium, which is the point at which no more unique knowledge (i.e., solutions) is being generated by agents in the networks.

We identify two types of diversity: ability and knowledge. Knowledge broadly speaking is an agent's knowledge of the relationship between its behavior and its performance. In our models below, we assume that there is no memory, so an agent's knowledge at time *t* is simply its solution and performance score at time *t*. Thus, in an NK space with an *N* of 5, an actor with a solution 01000 would simply know what performance was associated with that bit string. Ability is the capacity to convert knowledge into new knowledge. Different types of agents, given the same starting point, will have different capacities for creating new knowledge. Thus, one actor might be able to turn single 0s to 1s, and thus, from 01000 could evaluate the performance associated with 11000; 01100; and so on; and another actor might be able to turn single 1s to 0s. We discuss our formalization of these two types of diversity in turn below.

*Diversity of ability* is modeled by creating two "species" of agents[7] in the network: an A species agent with a distinct ability to search the problem space and a B species agent with a different, but also distinct, ability. A and B agents produce mutually exclusive sets of knowledge from the same starting point. (Here, our focus is on mutually exclusive capacities; see more about this in the robustness discussion and the Supplemental Methods and Supplementary Fig. 2 through 6.) All agents can only see a small portion of the problem space from a given starting point, given their limited search heuristic, which operates like a tiny "spotlight" on a vast terrain. While agents A and B cannot see the same areas of the problem space (i.e., non-overlapping spotlights), *how* agents A and B explore the problem space (or its search heuristic) is unique. For each problem space, agent A and agent B are each assigned a unique search heuristic that are mutually exclusive from the other. This is done to ensure that *how* agents explore any problem space is *not* driving these results (this is explained in detail in the "Methods" section). As such, referring to an agent as either "A" or "B" is just nomenclature: There is nothing inherent about the nature of an A agent or a B agent other than they both explore the problem space with mutually exclusive search heuristics. With this setup, when agents communicate with one another, one agent can lead the other agent to a better state of knowledge that they would have never been able to reach alone. In other words, agents can more readily find better states of knowledge than by communicating with similar agents (e.g., A with A or B with B)[7,8].

A and B agents are distributed in the network along a spectrum with two extremes, as shown in Fig. 2. In the case where there is minimal intermixing, all agents of the same species are adjacent

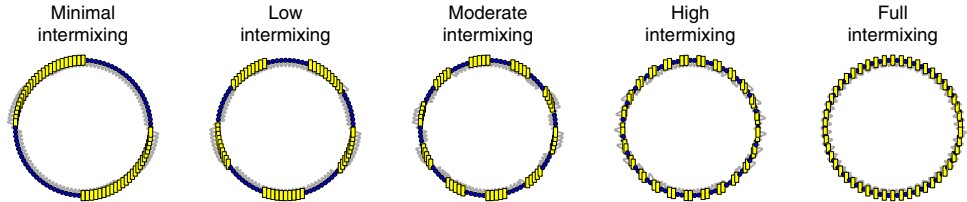

**Fig. 2** Intermixing of diverse abilities. Intermixing distributions of A and B agents in networks for the diversity of ability simulations

to each other, clustered into two large blocs of 50 agents. Here, different agents are almost fully segregated from one another, connected to other types of agents only at the edges of their respective clusters, at two points. We then alter the distribution of agents into blocs of 25 (low intermixing), 10 (moderate intermixing), and 5 (high intermixing). In the condition with full intermixing, the two types of agents simply alternate. For our torus network, agents are exposed to equal proportions of similar and different species of agents.

There are, however, multiple types of diversity (of ability) present in many real-life situations, such as multidisciplinary efforts at research universities that span several disciplines. Our model employs two types of agents at a time (e.g., A and B), as our focus is not on the *additive* benefits of additional forms of diversity[7–9], but instead on whether preserving local diversity reaps any systemic benefits. We can best highlight and isolate these effects by keeping the model simple enough to clearly underscore the interactions of two types of agents at a time. (For robustness, in the Supplemental Methods and Supplementary Fig. 1, we also test models that employ five types of agents set up in the exact same way as explained here. We find that our results still hold.)

We model *diversity of knowledge* by varying the distribution of the starting state of knowledge in the network. Whereas before we focused on agents with differing abilities, here we focus on agents' states of knowledge and assume agents have identical abilities. Recall that at the start of each simulation, agents are given initial starting states of knowledge. Juxtaposed to the diversity of ability setup, we distribute five initial states of knowledge (instead of 100), but to a population of 100 agents in the network (of just one species, instead of two). This species of agent is the same as the A/B configuration in the diversity of ability setups such that each simulation is assigned a unique search heuristic, to ensure that the trends are not driven by the choice in agents (but the same search heuristic is used across all levels of intermixing to compare across setups).

The initial states of knowledge are distributed in a similar fashion as the agents were distributed with the diversity of ability simulations, shown in Fig. 3. In the first configuration with *minimal* intermixing, five blocs of 20 adjacent agents are all assigned the same initial state of knowledge (i.e., 20 adjacent agents assigned to some state of knowledge $State_1$ and the next 20 adjacent agents assigned to another state of knowledge $State_2$, and so forth). Furthermore, initial states of knowledge are distributed in to 25 blocs of 4 agents with the same state of knowledge (low intermixing), 20 blocs of 5 agents with the same state of knowledge (moderate intermixing), 50 blocs of 2 adjacent agents (high intermixing). In each of these mixing setups, the five states of knowledge are randomly assigned to each of these blocs. For instance, the first bloc of agents may get $State_1$, but the next bloc of agents may get $State_3$, and the following bloc might get $State_5$, etc. This is because always having agents (or blocs of agents) with $State_1$ always next to $State_2$, or $State_4$ always next to $State_5$, (and so forth) will negate opportunities for agents with $State_1$ to interact with agents with $State_4$ or $State_5$, and vice-versa. Thus, randomization allows for multiple pathways for agents to explore.

Finally, for the last setup, the five states of knowledge are randomly distributed in equal proportions across the 100 agents (random intermixing).

## Results

**Analytic setup**. For our baseline results presented here, we run 10,000 simulations, each with a distinct NK problem space and a simple torus network of agents with no contagion effect. We group our findings into two major results. For the diversity of ability models, systemic performance improves with increased intermixing and these results are consistent for all time scales. The simulations were run so that each of the five levels of intermixing had identical starting points—for 10,000 starting points. We could thus compare how well each level of intermixing did by comparing their average NK scores across agents for each network and then comparing these average network performances across NK problem spaces.

**Diversity of ability**. Figure 4a summarizes the results for the diversity of ability simulations in NK spaces. The y-axis measures the average NK score across problem spaces for each intermixing setup, which are ordered on the x-axis from minimal intermixing on the left to full intermixing on the right. In other words, each of these setups are given the exact same problem space and starting solutions. The only thing that differs among them is how the A and B agents are distributed (i.e., intermixing rate). As soon as each of the five setups reach an equilibrium state (i.e., no more new solutions are being introduced), their final scores are averaged across all of the agents in the network for each round in the simulation. (In the case of equilibrium, all agents have the same solution and, thus, scores.) These values are then averaged across all problem spaces for each intermixing setup. This is represented by the dots and the different colored plots corresponding to rates of *intermixing* in Fig. 4. The dots in 4 are the average scores for each network setup across 10,000 NK problem spaces. As such, we also include the standard errors from these spaces as error bars in this figure and all subsequent figures. Since Fig. 4a only shows the final average result of these simulation runs, Fig. 4b plots these average NK scores in the same way, but over time, from start to equilibrium (along with the standard errors just as in Fig. 4a).

Out of 10,000 NK spaces, networks with minimal intermixing had an average NK score of 0.47, while networks that were fully intermixed had an average of 0.95. We find that each increment of intermixing is associated with improved performance. As intermixing increases, the overall performance of the system improves monotonically. Why does intermixing yield better systemic results and why does this hold across time scales? Prior research highlights the benefits of *maintaining* diversity of ability in group deliberation, and we find a similar pattern here: the more locally diverse systems are (i.e., the more intermixing), the greater the number of unique states of knowledge generated. That is, if we imagine an actor of type A that finds a good solution, an adjacent actor of type B can copy that solution and build on it. If there is no actor B nearby, then that good solution may be

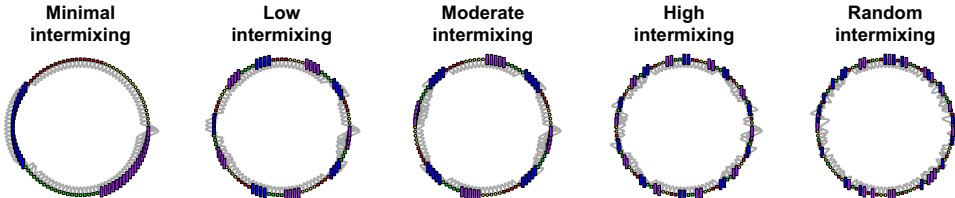

**Fig. 3** Intermixing of diverse knowledge. Intermixing distributions of the five initial states of knowledge in networks for the diversity of knowledge simulations

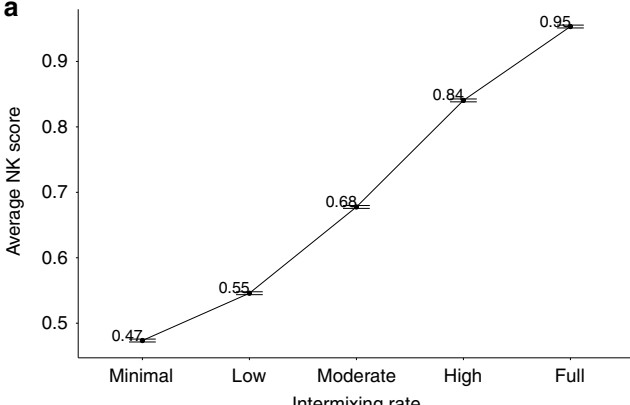

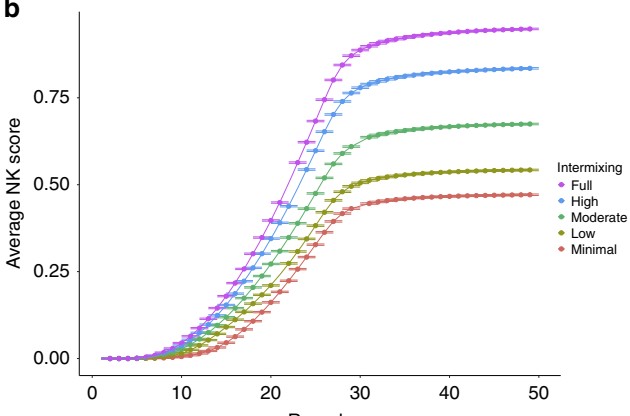

**Fig. 4** Diverse ability intermixing performance. **a** The average NK score of networks across problem spaces, plotted by intermixing rates for the diversity of ability simulations. **b** The average NK score of networks across problem spaces, plotted over time for the diversity of ability simulations by intermixing rates. Error bars are the standard errors measured across NK spaces

replaced by a "better" solution that is a local optimum for both A and B (and thus a dead end for further innovation).

Figure 5 plots the average number of unique states of knowledge generated by the five distributions in the NK spaces. For ease of presentation, we transform these plots such that they are the marginal performance over the minimal intermixed system (hence, the minimal intermixing networks is set to zero, and the other trends are the number of unique solutions subtracted by the minimal intermixing network trend). In other words, the more intermixing there is in a system, the more unique solutions are generated, here measured relative to the number of unique solutions generated by the minimal intermixing networks. In short, the more intermixing, the more knowledge the system produces. As Page[9] notes, "Each of us walks around carrying a

toolbox filled with a variety of tools" (p.103), that "enable collections of people to find more and better [states of knowledge]." (p.13) However, this work focused on small groups with plenary discussion and varying group composition[7,8], but no variation in network configuration, as is explored here. Indeed, this result extends his collective work[7–9] to a network of problem solvers, illuminating that it is important to maximize the number of tools available to apply to a given solution in order to maximize the number of new states of knowledge that can, in turn, be generated. The way to maximize the number of tools available for each state of knowledge generated in a system is to have the maximum possible intermixing of the abilities of different actors.

Intermixing with respect to initial knowledge states shows a dramatically different pattern. For the diversity of knowledge, there is a tradeoff of systemic performance in time: intermixing improves short-run but hurts long-run performance.

**Diversity of knowledge**. Figure 6 refers to our diversity of knowledge results in the NK problem space. These results parallel our presentation of the diversity of ability results. Like the previous Fig. 4a, Figure 6a measures the average NK score found by agents in the network in a particular round for a specific NK problem space. These scores are then averaged across NK spaces for each of the five intermixing setups, represented by the dots and the different colored plots corresponding to rates of intermixing in the same way as they were setup for Fig. 4a, b. (And the standard errors reflect the averages of networks across the NK spaces, just as before.) Similarly, Fig. 6b plots this performance over time, from start to equilibrium.

Here, however, we find quite different results: intermixing *hampers* systemic performance. Furthermore, there is a tradeoff in systemic performance over time, as seen in Fig. 6b. Networks with some form of intermixing perform better in the short-run. However, the network with the minimal possible intermixing performs worse in the short-run but best in the long-run. In other words, this suggests an "all or nothing" trade-off over time: the configuration with the least amount of intermixing possible (minimal) performs worse in the short-run, but better in the long-run, while any gradation of increased intermixing yields the same rank-ordering in performance as their counterpart networks in the diversity of ability simulations.

Agents in the intermixed network setups explore for solutions early on in the simulation, which is a double-edged sword. In the short-run, with more intermixing, agents quickly turn to *exploitation*, as they merely take solutions that are marginally better relative to other setups but are still mediocre in *absolute* terms. Said differently, agents in setups with at least some intermixing quickly coalesce to whatever the few agents that did explore the problem space found, which are often not the best solutions possible. So, exploring the problem space using worse solutions often leads to modest gains. However, the agents in setups with minimal intermixing are more commonly *exploring*, rather than *exploiting*, because their neighbors' initial solutions are less optimal than their counterparts in intermixed networks.

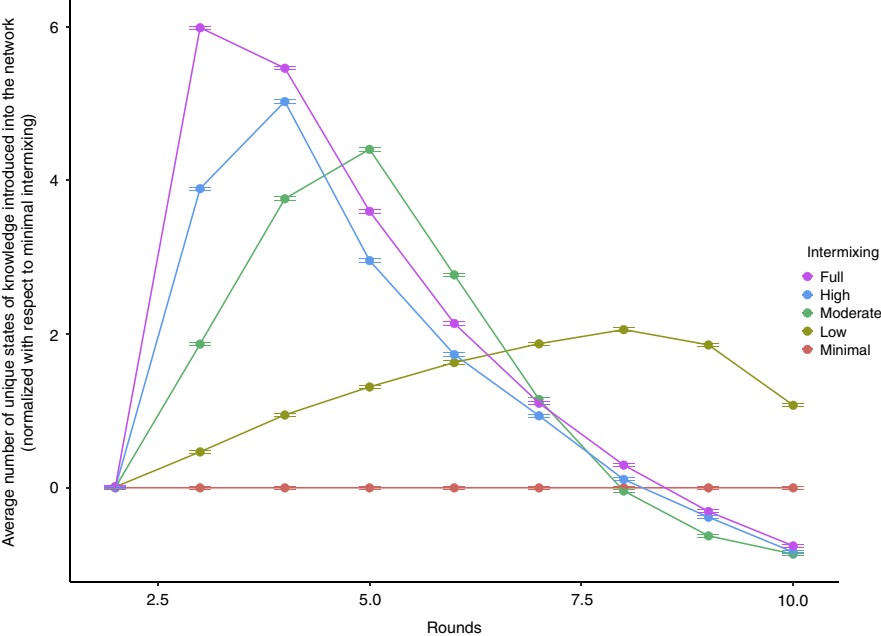

**Fig. 5** Unique knowledge produced by diverse ability intermixing. The average marginal number of unique states of knowledge over time for the diversity of ability simulations—as compared to the number of unique solutions generated by minimal intermixing networks—by intermixing rate. Error bars are the standard errors measured across NK spaces

This is because there is minimal exposure to diversity in these *minimal* setups, which inadvertently produces poor solutions in the short-run but allows for more *exploration* to find better solutions in the long-run. In other words, by finding better solutions through *exploration*, agents more often uncover pathways to better solutions earlier on. The end result is that the long-run performance of setups with minimal intermixing is best.

Figure 7 sheds some light as to why little intermixing leads to better long-run systemic results across NK spaces. For each of the intermixing setups, we calculate the average number of unique states of knowledge (i.e., solutions) introduced into the network for each round. Figure 7 plots this for each of the five intermixing scenarios, and they are measured with respect to random intermixing, the setup that performs the worst in the long-run. (Hence, random intermixing is set to zero, and the other trends are measured with respect to it.) Networks with low to random intermixing introduce fewer unique solutions, on average, when compared to networks with minimal intermixing. In so doing, networks with minimal intermixing buffer promising but initially inferior solutions from being wiped out from the system prematurely, preserving alternative pathways to better states of knowledge. By preserving these initially inferior solutions, the system takes a hit in performance in the short-run, but maintains more pathways into the solution landscape, allowing more exploration, and thus greater performance, in the long-run.

## Discussion

In sum, we find that more intermixing monotonically increases systemic performance across time for diversity of ability. However, for diversity of knowledge, intermixing results in higher levels of performance only in the short run, as minimal intermixing preserves systemic diversity in the long-run. (In the Supplemental Methods and Supplementary Fig. 7 through 11, we show that these results hold in other problem spaces, contagions, and network structures.)

These findings come with particular scope conditions, that derive from the model's assumptions. They are relevant to optimization problems that individuals (imperfectly) solve, typically getting stuck in local optima, but where those individuals can learn from others. Individuals are myopic—i.e., able to produce solutions within a small neighborhood around the current solution. All individuals are attempting to solve the identical problem; but individuals vary in their position in the network, and in their ability to produce new solutions given their knowledge. This potentially encompasses a broad set of human problem solving. Examples of social learning that approximately fit into this paradigm might include superintendents of high school districts (as discussed above); graduate students trying to succeed in their program; or small-business franchises. In each case, the relevant agents are trying to solve a complex problem, and can learn from the behavior of other actors, with whom they are generally not in direct competition with. However, our approach excludes a broad set of collective problem behavior; e.g., where skill in groups is coupled with specialization of roles. These results have little implication, for example, for the success of a baseball team; or for scenarios where agents are solving substantially different problems (e.g., career success in very different fields).

As such, this work is meant to be a platform for further research, both on the modeling side and with behavioral research. For example, there are other mechanisms through which diversity may be maintained[33,56–58]. Actors may have exogenous drivers of attitudes[56,57,59]; successful spreading may require multiple, reinforcing signals[58]; and individuals may simply be stubborn. Actors may also have better answers to sub-problems, which might increase the amount of "knowledge recombination" that occurs. Future work also ought to offer agents opportunities for search capabilities that are learned from other agents, as people learn skills and perspectives from others who are different to them (e.g., Burt[41]). Finally, an agent ought to be able to exchange ties with the chance that these newer ties might connect with better performing agents.

Future research should integrate behavioral research, both experimentally in lab settings (by varying the network structure of problem solving groups[17,33,60]), and in field settings. This can

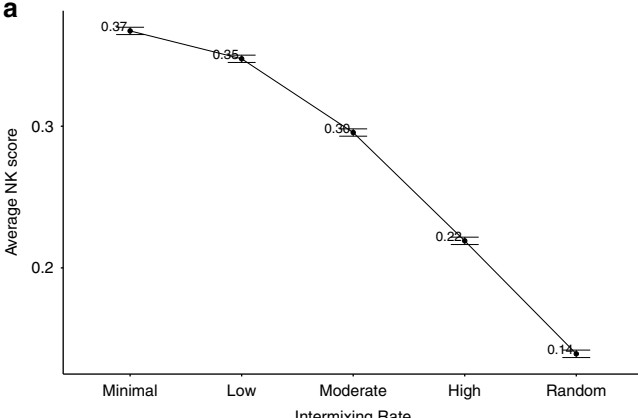

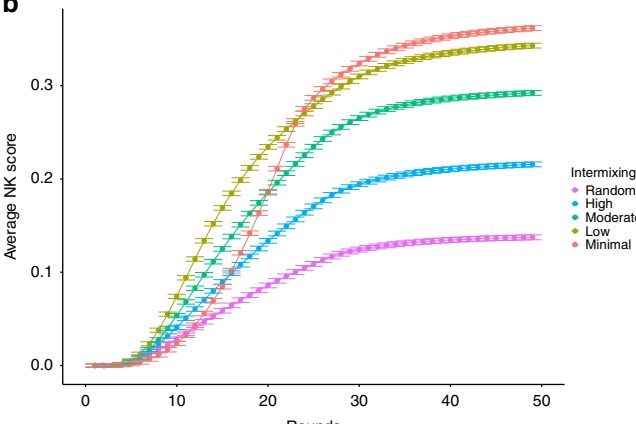

**Fig. 6** Diverse knowledge intermixing performance. **a** The average NK score of networks across problem spaces, plotted by intermixing rates for the diversity of knowledge simulations. **b** The average NK score of networks across problem spaces, plotted over time for the diversity of knowledge simulations by intermixing rates. Error bars are the standard errors measured across NK spaces

evaluate the micro-behavioral assumptions (e.g., around social learning), test the propositions regarding emergent behaviors, examine whether there are countervailing behavioral effects (e.g., whether local diversity reduces prosocial behavior[61,62]), and perhaps iterate in evaluation of alternative specifications for further ABMs.

More critically, this work is meant to shed light on how the distribution of diversity affects deliberative problem-solving across multiple scales and social collectives, ranging from small groups to modern democracies. Given that research on how people tend to organize themselves is strongly biased towards homogeneity, we need to consider multiple levers to facilitate or inhibit the distribution of diverse individuals and ideas. To that end, our findings highlight the potential value of creating loci for multidisciplinary collaboration in settings such as universities and knowledge-creating companies. However, the flip side is that there may be value in creating an array of "hot houses" of exploration by likeminded sets of individuals, temporarily sheltered from systemic forces of conformity.

## Methods

**Experimental setup**. All of our models were written and executed in Python 2.7, a high-level object-oriented programming language. The NK problem spaces were generated as text files that contained the address of every point in the space

(sequence of 1s and 0s of length $N$ that identifies every point in the NK space) and its score for the tuning parameter set to $K = 5$.

A simulation is comprised up of a torus network of 100 agents that completes after some number of rounds when all of the agents reach equilibrium (i.e., they are not generating any new unique solutions). A simulation begins with a problem space for agents to explore. Each agent is given some state of initial knowledge (e.g., a random sequence of 1s and 0s of length $N$). For robustness, agents start with the same sequence of 1s and 0s in each simulation. However, since each problem space is completely distinct and different for all other problem spaces, there is nothing intrinsically good or bad about any of these solutions when used in different problem spaces. In other words, a solution in one problem space may be exceptionally good, but very bad in another. (Changing an agent's initial solutions for each solution results in no discernable change in any of our results.).

We do hold our solution and problem space setup constant across our various treatments (i.e., intermixing of diversity), and across our robustness checks (i.e., network structure, contagion, etc. that we outline in our Supplemental Methods and Supplementary Fig. 2 through 11). Thus, our setups are identical in every way *except* for how we intermix diversity in the network, along with our robustness checks.

**Problem spaces**. An NK space is operationalized by an $N$ dimensional sequence of 0s and 1s, where the marginal contribution to performance of a given 1 (or 0) is contingent on $K$ other bits (where each contingency is generated by a uniform distribution between 0 and 1). The 1s may be viewed as the presence of an activity, and the 0s its absence. If a cluster of activities, together, produces a notable improvement in performance that is greater than the sum of each of these activities in isolation, these activities may be viewed as synergistic. The higher $K$ for a given $N$, the more local optima there will be in an NK space. We used the parameter values of $N = 20$ and $K = 5$ for our simulations and all NK scores are normalized to each space's global maximum. Hence, an NK score of 1.0 is the maximum that can be obtained for any given NK space. As with Lazer and Friedman[39], a monotonic adjustment is made to the score by raising the raw score produced in the standard NK model to the 10th power and then divided by a constant so that the transformed values are between 0 and 1, yielding a distribution where the model random solution is quite poor relative to the optimum.

Every agent is associated with a unique array of length $N$, called a[$i$], as show in Fig. 8. Each entry in the array contains either a 0 or a 1. The specific sequence of 0s and 1s will ultimately yield a normalized and cumulative score, or NK score, for this particular array. Each of the $N$ entries in the array is associated with one of several potential scores, a decimal value between 0 and 1, exclusively. The determination of this score depends on two things: (1) whether the entry in the array has a value of 0 or 1, and (2) whether $K$ other entries in the array have the values 0 or 1. These $K$ dependencies are unique and different for each of the $N$ entries. To illustrate this point, consider the fifth entry in this hypothetical array show in Fig. 8, which, for simplicity, we will say is dependent on $K = 2$ other entries: say, the second and sixteenth values in the array, for instance. Each entry can only take on a value of a 0 or a 1, so there are a total number of 8 combinations of 0s and 1s, because each entry that depends on the values of $K$ other entries in the array can potentially represent $2^{K+1}$ scores. Each of these eight combinations is associated with a particular decimal score between 0 and 1, exclusively. So, if the second, fifth, and sixteenth entries all had 0, then the fifth entry would be associated with some unique score (i.e., 0.15 as per Fig. 8). If the second entry were instead a 1, then the fifth entry would be associated with a different unique score, and so forth. The scores for each entry in a[$i$] are iteratively calculated averaged together to reflect the unique score for this particular string of 0s and 1s. In other words, the resulting product is the NK score associated with this agent's particular array (i.e., state of knowledge).

**Modeling diversity**. There are different ways of envisioning diversity of ability. Different types of agents may have non-overlapping (i.e., mutually exclusive) search capabilities, meaning that when agents are given the same state of knowledge, the new knowledge that they can generate is completely different. Or, they might have a much larger search capability than other agents and can generate the knowledge that other agents can, plus additional sets of knowledge. Hong and Page[7] model diverse types of agents based on having different perspectives of their problem space and heuristics for manipulating potential sets of knowledge (i.e., solutions) while exploring the space. The interactions of these distinct pairings of agents might still lead to the same state of knowledge, but more often than not, they open pathways to optimal states of knowledge that agents could not have found alone. Here, our focus is on mutually exclusive capacities (see more about this in the Supplementary Methods and Supplementary Fig. 2 through 6).

We model the agents that differ in ability as two species of agents—the A species and the B species—with two non-overlapping methods for generating new solutions. Agents explore a problem space by manipulating a state of knowledge with a specific heuristic unique to that species of agent. Here, we focus on how agents explore the NK problem space. Recall that a state of knowledge in the NK space is a bit array of 0s and 1s of length $N = 20$ that refers to the address of a point in the NK space holding a particular score. (For instance, an example is "1001100011001010100" refers to some score between 0 and 1 inclusive, where 1 refers to the highest possible score in any particular space.)

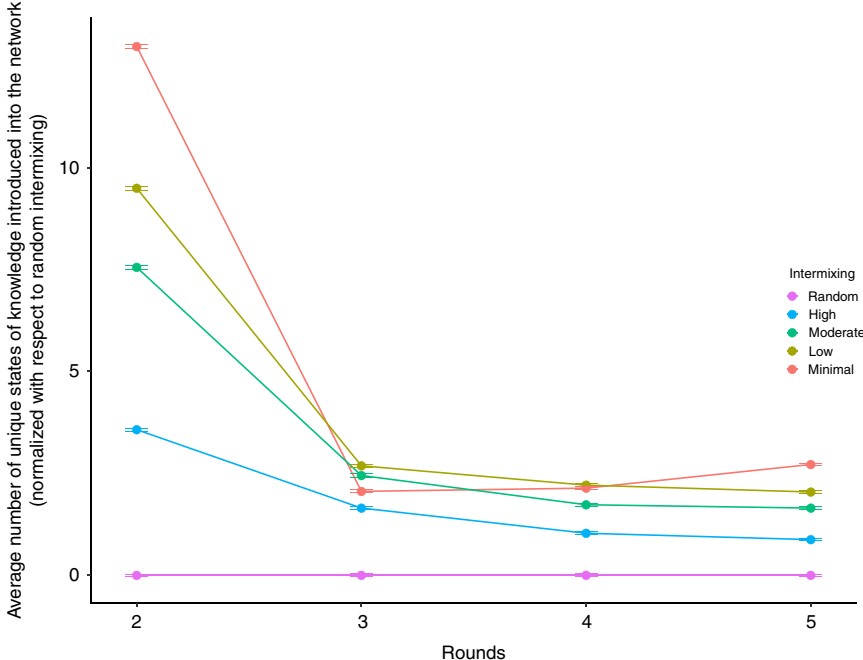

**Fig. 7** Unique knowledge produced by diverse knowledge intermixing. The average marginal number of unique states of knowledge over time for the diversity of knowledge simulations—as compared to the number of unique solutions generated by random intermixing networks—by intermixing rate. Error bars are the standard errors measured across NK spaces

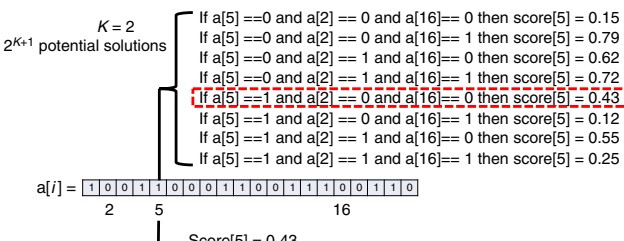

**Fig. 8** Calculating NK scores. How NK scores are calculated using hypothetical values for a solution using $N$ and $K$

Each simulation generates a unique form of ability diversity such that A and B ought to have mostly non-overlapping search capabilities. This ensures that the results are not the result of one particular type of heuristic that any one agent engages in. Thus, referring to agents as either "A" or "B" is merely nomenclature to distinguish them, rather than anything that is inherent about the quality of being an "A" agent or a "B" agent.

The agents are randomly assigned a manipulation mask (or *mask* for short) for each problem space (and held constant cross all five intermixing setups). The mask is simply the unique bits in the NK string of 1s and 0s that it will manipulate. One agent is randomly assigned anywhere from 2 to $N$-2 random bits that it can manipulate, while the other agent is randomly assigned anywhere from 2 to some number of bits less that is than the number assigned to the other agent. For instance, the A agent may be assigned bits 3-5-10-18 (where 3 refers to the third bit in the sequence of 1s and 0s; 5 refers to the fifth bit; etc.) while the B agent may be assigned 6-13-14-15-16-19. The only strict criterion is that A and B agents can never use the same mask at the same time. Similarly, agents are then randomly assigned one of four manipulation behaviors for each problem space and these selections are held constant across all five intermixing setups. Again, the only strict criterion is that A and B agents can never the same manipulation behavior at the same time. In the NK space, agents can (1) turn every bit in its mask to a 1; (2) turn every bit in its mask to a 0; (3) randomly jumble all of the bits in its mask; or (4) turn every bit in its mask to its opposite, or from 1 to 0 and 0 to 1.

To ensure that agents do not easily become stuck at local optima, agents explore the NK problem space by "shifting" their mask in a step-wise fashion (one bit at a time) and apply their manipulating behavior to each of the $N = 20$ "shifted mask" bits. And out of the $N = 20$ possible mutations, the agent adopts the best state of knowledge. For instance, consider an agent who has the mask above (3-5-10-18) and turns each bit in this mask to a 1. So the agent takes the state of knowledge

given by 10011100110011010100 and converts this to 10111100110011010100. (The third bit which was a 0 becomes a 1, while the fifth, tenth, and eighteenth bits which were all 1s remain 1.) After the score associated with this particular manipulation is calculated, the agent then shifts the mask by one (3-5-10-18 becomes 4-6-11-19). This shifted mask (using the same behavior of turning each bit to 1) is applied to the original state of knowledge (10011100110011010100) and turns it to 10011100111011010110. (The fourth and sixth bits remains a 1, while the sixth and eleventh bits change from a 0 to a 1.) This process continues until 20 manipulations are calculated (and their associated scores for each step) through the $N = 20$ bits. (When the bit in the shifted mask is greater than $N$, the bit cycles back to the beginning, such that the "21st" bit is really the 1st bit in the state of knowledge, and so forth.) Finally, the agent adopts the best state of knowledge out of the 20 possible setups it can possibly create after altering the original state of knowledge.

**Robustness**. We evaluate the robustness of our findings by rerunning all of our experiments varying several dimensions: (1) the problem space the agents explore; (2) the information contagion that dictates the rate of information diffusion in the network; (3) and the network structure that preserves intermixing. Given the large array of permutations of computational experiments we ran, and for simplicity of presentation, we present our robustness checks in the Supplementary Methods but only directly reference our main results from our baseline condition (i.e., a simple torus network with no contagion effect in the NK problem space with A and B agents) in the results section. However, we note that the qualitative pattern of our findings is notably robust across this wide-array of model specifications.

**Reporting summary**. Further information on research design is available in the Nature Research Reporting Summary linked to this article.

## Data availability

The code used to generate the data are available on Harvard's Dataverse: https://doi.org/10.7910/DVN/8ZXFHI.

## Code availability

All code used to run these simulations are available on Harvard's Dataverse: https://doi.org/10.7910/DVN/8ZXFHI.

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

## Acknowledgements

This research is based upon work supported in part by the Office of the Director of National Intelligence (ODNI), Intelligence Advanced Research Projects Activity (IARPA) under contract 2017-17061500006. This research was supported by Office of Naval Research Grant G00005072. The views and conclusions contained herein are those of the authors and should not be interpreted as necessarily representing the official policies, either expressed or implied, of ODNI, IARPA, ONR, or the U.S. Government. The U.S.

Government is authorized to reproduce and distribute reprints for governmental purposes notwithstanding any copyright annotation therein.

## Author contributions

Both authors contributed equally to this work. D.L. designed the original setup for the experiments. C.G. implemented the experiments in Python and created the figures with R. Both D.L. and C.G. designed updates to the original experiment setup and wrote the manuscript.

## Competing interests

The authors declare no competing interests.
