## [Peer Review File · Nature Communications]

Reviewers' comments:

Reviewer #1 (Remarks to the Author):

The paper shows that integration of different abilities improves performance but that integration of knowledge can have detrimental long-term consequences. I find this an interesting finding that would be valuable to those that study the evolution of human behaviour and social change, but I don't have the expertise to know if this is novel in other relevant fields: presumably fields looking at machine learning algorithms and search algorithms may have similar results? Have others in computational social science looked at this?

The authors indicate that their model applies to collective problem-solving (they need to remove a 'collective' in their sentence), but it is clear that humans don't learn and recall information in anything resembling different switching states of a binary string. While I think this form of abstraction is probably justifiable, the case needs to be made.

In the discussion, the authors go so far as to suggest forms of social change as a result of their findings ("Specifically, we should find ways to encourage people with different abilities salient to complex problem-solving to interact with one another."). The authors should hold off making recommendations for policy change based only on the results of an abstract model - the effect has not yet been tested in experimental contexts or using real-world data, nor have the social consequences of such recommendations beyond knowledge generation been considered.

The analysis appears to be fairly comprehensive, in that they have covered a variety of scenarios in the supplementary information.

Other comments:

It is not clear why, when investigating the effects of knowledge distribution, they only have two configurations (minimal and full), rather than a variety (equivalent to the diversity of ability investigation).

As someone that is not familiar with all the NK search space literature, I find the description of Fig. 1 vertical axis, and equivalent figures confusing. The language used is different in the axis legend, the Figure legend and the main text. Having read it, I still don't really understand what it means to 'win'. I also remain unclear about the relationship between the variation in binary sequences that can be explored and the fitness consequences. Perhaps the authors can use an example (as they do for other parts of the process) which illustrates the mapping between sequence and fitness score. The sentence "It measures the percentage of problem spaces in which any given configuration found the best state of knowledge amongst the five, which also allows for ties" indicates that there are only five different fitness scores. If correct, there must be a lot of redundancy across possible sequences...as you can see, readers similar to me may not fully understand.

The graphs often show the lines for the mean effects, but it would be good to have a representation of variation across runs: it is important to know if the mean trends are clouded by large variation.

Reviewer #2 (Remarks to the Author):

The authors developed an agent-based model to investigate the role of diversity in ability and knowledge in solving problems in social systems. To do this, agents solved problems as groups on rugged Kauffman type NK landscapes. They found diversity in ability greatly aided problem solving but not diversity in knowledge. Their approach is very general and interesting, but there are

several things that need clarification and untangling, which I will list below:

1. There are two species of agent described in the methods section with two non-overlapping methods for generating new solutions. Are these agents that differ in ability? What are agents that differ in knowledge?
2. The specific problems agents solve are not specified in the manuscript. Solutions are strings of "1s" and "0s" of length 20, but how are they introduced? How do agents evaluate the possible solutions they generate?
3. How do other strategies for generating possible solutions do? In the supplementary material, an A+ species is discussed, which flips bits in a string. Why is it only discussed in the supplementary material? If it produces the union of strategies A and B, shouldn't homogenous groups of A+ species do as well as diverse groups of A and B?
4. What does it mean to come in "first place", which is in the y-axis label of many figures without definition in the text?
5. What computer language and/or agent-based simulation environment was used?
6. Overall, the description of the model and simulation process is too unclear. For example, on page 20, lines 273 to 275: "Experimental Setup. Each simulation begins with a new problem space for agents to explore, but with the same state of initial knowledge given to the agents as the previous simulation (i.e., same solutions used in the previous simulations.)" How is the previous simulation setup?

Reviewer #3 (Remarks to the Author):

Using ABM this paper examines whether clustering the diversity of knowledge or skills within a system leads to better performance than dispersing the same quantities of diversity. Basically, knowledge diversity involves agents having different knowledge but the same skills and skill diversity involves agents having different skills but same knowledge. At a high level of abstraction, the authors measure how varying the percentage of skill diversity from low-to-high corresponds with changes in the fraction of time a percentage mix of diversity (e.g, no diversity, low, medium, or high) came in first place in finding the best answer. A similar manipulation is done with knowledge diversity. The authors find that a positive relationship between intergroup mixing rate and performance (mixing skills is good) but the opposite for knowledge diversity (mixing knowledge is bad). The simulation is done competently and the supporting material is thorough and detailed (albeit very long and somewhat daunting to follow).

I have several comments for the authors.

First, this is a great problem to solve and has lots of potential applications, especially in providing useful rules of thumb on organizational design. However, I found myself incapable of identifying applications. The key reason is the level of abstraction of the paper makes the findings nearly uninterpretable and impossible to confidently to apply to a real life situation. Because the ABM and manipulations are done without any specification of context, it was possible to think of any example that could be both consistent and inconsistent with your model's findings. For example, let's say there are farmers and ranchers, two groups that differ in skills. A rancher knows that killing wolf packs means less cattle lost to predation. So, other ranchers copy the practice and so do farmers who want to help the ranchers and live on lands in between grazing land. So, if performance is measured as fewer dead cattle then mixing is trivially good because wolves are exterminated quicker and more completely. But, say change the situation is examined from a

slightly different perspective, once all the wolves are exterminated, there is an explosion in the deer and buffalo population, which means cattle have less grazing ground and farmers are having their crops eaten. The explosion of deer also brings ticks, which make the ranchers and farmers sick and unable to work, and the general public's health care bills are rising, leading to even greater rather than less costs. So, by arbitrarily specifying a situation, the model can be shown to fit anything, which makes the model unable explain very much in the end. I suppose the counter argument to my criticism that you can't make everyone happy. Nevertheless, if all that you want to improve is the welfare of the ranchers, then I think the problem is pretty easy and you don't need a model to explain it.

Second, if the model is applied to real-life situations where skills differences can be measured, it doesn't explain many common phenomena. Modern organizations always cluster persons together of similar background and cognitions. This is why organizations have departments, units, subunits and so on. So, how generalizable is your finding that increasing mixing levels of skill diversity is good? Doesn't the level of mixing depend on many other factors, including coordination needs, redundancy needs, demand, and other visible factors that your model ignores. Some real-life illustrations would have helped move the paper from a mechanical treatment of a very hard problem (which is why it has not been solved) to a sophisticated analysis.

Third, the model focuses on a toy situation where you have just A and B. But in most real-life situation there are A, B, C, and D types of diversity. We have the Democrats and Republicans but Independents play an important role too.

Reviewer 1

1). The paper shows that integration of different abilities improves performance but that integration of knowledge can have detrimental long-term consequences. I find this an interesting finding that would be valuable to those that study the evolution of human behaviour and social change, but I don't have the expertise to know if this is novel in other relevant fields: presumably fields looking at machine learning algorithms and search algorithms may have similar results? Have others in computational social science looked at this?

The role of diversity within groups has been substantially explored in the social sciences, most prominently in recent years by Scott Page and collaborators (but by many others as well). However, as per the reviewer's intuition, researchers in computer science, genetics, and operations research have explored diversity in learning and search algorithms, but, to the best of our knowledge, have yet to test how the distribution of diversity shapes overall outcomes. Most of their efforts are focused on how maintaining a certain level of diversity within some population shapes overall outcomes, for example, see:

*Nsakanda, Aaron Luntala, Wilson L. Price, Moustapha Diaby, and Marc Gravel. 2007. "Ensuring Population Diversity in Genetic Algorithms: A Technical Note with Application to the Cell Formation Problem." *European Journal of Operational Research* 178(2):634–38.*

*Watson, Tim and Peter Messer. 2001. "Increasing Diversity in Genetic Algorithms." Pp. 116–23 in *Developments in Soft Computing, Advances in Soft Computing, Physica, Heidelberg.**

*Yuan, L., M. Li, and J. Li. 2008. "Research on Diversity Measure of Niche Genetic Algorithm." Pp. 47–50 in *2008 Second International Conference on Genetic and Evolutionary Computing.**

We have refined our literature review/conclusion to make clear our value added over the existing literature.

2). The authors indicate that their model applies to collective problem-solving (they need to remove a 'collective' in their sentence), but it is clear that humans don't learn and recall information in anything resembling different switching states of a binary string. While I think this form of abstraction is probably justifiable, the case needs to be made.

We agree that a key element of any modeling paper is whether the underlying abstraction--the metaphor that the model somehow captures essential elements of how the world works--is actually persuasive. We have therefore rewritten and sharpened the section of the paper discussing the motivation of the model. We note also that this type of abstraction is fairly common in the modeling literature, as cited in our paper.

We also agree with the point that humans "don't learn and recall information" in any fashion that reflects the verbatim processes in our model. Our essential argument is that we imagine a solution that someone comes up with as

comprised of a “basket of activities.” These activities, in turn, have synergies amongst them. So, in our setup, a “1” represents the presence of an activity, and a “synergy” exists when the contribution of two activities together exceeds the sum of the contribution of each activity in isolation. The NK space we use captured the interaction of activities and these synergies. Having a limited ability to switch bits is meant to capture the stylized fact that humans have very limited ability to search problem spaces relative to their scale. We note that there is substantial precedent in the literature for this sort of formalism, based on a similar motivation. Furthermore, the fact that we use a dramatically different abstraction (i.e., the Travelling Salesperson Problem space) to also represent decision-making to test the robustness of our results is meant to underscore that our results are not an artifact of our NK abstraction. This in paper replication with a different problem space, in fact, goes beyond the standards found in the literature.

3). In the discussion, the authors go so far as to suggest forms of social change as a result of their findings ("Specifically, we should find ways to encourage people with different abilities salient to complex problem-solving to interact with one another."). The authors should hold off making recommendations for policy change based only on the results of an abstract model - the effect has not yet been tested in experimental contexts or using real-world data, nor have the social consequences of such recommendations beyond knowledge generation been considered.

We agree that no one should make policy changes based on our paper (and very rarely ever based on a single paper in any case). That said, discussing possible actionable implications of findings is important motivation-- these findings, if they hold up to subsequent scrutiny, matter for organizations and for society more generally. We have thus softened this discussion significantly, by suggesting that if empirically validated, these results have actionable implications.

4). It is not clear why, when investigating the effects of knowledge distribution, they only have two configurations (minimal and full), rather than a variety (equivalent to the diversity of ability investigation).

Excellent point; there really was not a good motivation for the limited array of “treatment” conditions when examining knowledge distribution. To that end, we have redone our analyses to include a gradation in configurations for the diversity of knowledge simulations, from no intergroup mixing to random intergroup mixing. This mirrors what we did for diversity of ability simulations. We show in the main text that our findings still hold. The only difference is that we randomly assign solutions to each bloc and our full intermixing setup is a random assignment of the five states of knowledge across the 100 agents. When dealing with more than $n=2$ solutions, this becomes an issue, as keeping solution 2s always next to 1s and 3s might miss the opportunities to instead be next to 4s and 5s (and so forth). Hence, we randomize these solutions to reflect full intermixing.

5). As someone that is not familiar with all the NK search space literature, I find the description of Fig. 1 vertical axis, and equivalent figures confusing. The language used is different in the axis legend, the Figure legend and the main text. Having read it, I still don't really understand what it means to 'win'. I also remain unclear about the relationship between the variation in binary

sequences that can be explored and the fitness consequences. Perhaps the authors can use an example (as they do for other parts of the process) which illustrates the mapping between sequence and fitness score. The sentence "It measures the percentage of problem spaces in which any given configuration found the best state of knowledge amongst the five, which also allows for ties" indicates that there are only five different fitness scores. If correct, there must be a lot of redundancy across possible sequences...as you can see, readers similar to me may not fully understand.

The graphs often show the lines for the mean effects, but it would be good to have a representation of variation across runs: it is important to know if the mean trends are clouded by large variation.

The language around the NK space was confusing, as was our discussion of our measurement of each configuration's performance. Instead, in the main text, we report the performance of each network setup in terms of its score in the NK space, which are each normalized such that the maximum score is 1 and the minimum possible score is 0.

In any case, in the manuscript, we have:

- (1) clarified the explanation of what it means to "win",*
- (2) supplied an illustration to explain the relationship between the binary sequences and fitness.*
- (3) where appropriate, added a characterization of the variation across runs, by including the standard errors for most of the figures. (It does not make sense, for example, for the "percent win" figures in the Supplemental Section.)*

Reviewer 2

1). If the model is applied to real-life situations where skills differences can be measured, it doesn't explain many common phenomena. Modern organizations always cluster persons together of similar background and cognitions. This is why organizations have departments, units, subunits and so on. So, how generalizable is your finding that increasing mixing levels of skill diversity is good? Doesn't the level of mixing depend on many other factors, including coordination needs, redundancy needs, demand, and other visible factors that your model ignores. Some real-life illustrations would have helped move the paper from a mechanical treatment of a very hard problem (which is why it has not been solved) to a sophisticated analysis.

The reviewer brings up a critical point, one that cuts directly to what we are trying to accomplish here. As with most research, modelling the empirical world hone in on one element of some phenomenon: in our case, the distribution of local diversity in parallel problem solving. In particular for agent-based models, they simplify many social, economic,

and political forces normally at play and bring to the foreground a single point by metaphor. That being said, such approaches can yield much explanatory power. For instance, consider Thomas Schelling's segregation model. If one would interpret his model as fully capturing how systems work in the "real" world, then the policy and social interpretation is that the addition of one more different neighbor causes someone to move out of their neighborhood. In reality, many latent and explicit factors affect segregation. Further, there are many factors other than ethnic composition of their immediate neighbors that would affect the moving decisions of an individual. However, there is a lot of explanatory power in this highly simplified model of how segregation may be driven endogenously. (Indeed, how the preservation of diversity endogenously improves the performance of the system is one of our focal insights.) To that end, we agree that we did not do a good enough of a job motivating our model. We added a few examples in the introduction and the "Complex Problems" section that, while not perfect analogies that entirely map to every real-world example, reflect our model's overarching explanatory aim.

So: we acknowledge the point that there are other important factors with respect to organizational design. We seek to offer a "first principles" glimpse into how exogenously determined diversity might affect collective problem solving. In the discussion, we do call for future work to consider agents who self-select as to whom they connect with based on diversity. However, our focus here is to create a baseline for future work to explore these additional complexities.

2). Overall, the description of the model and simulation process is too unclear. For example, on page 20, lines 273 to 275: "Experimental Setup. Each simulation begins with a new problem space for agents to explore, but with the same state of initial knowledge given to the agents as the previous simulation (i.e., same solutions used in the previous simulations.)" How is the previous simulation setup?

The presentation of the model and simulation was imprecise at multiple points, and we have gone through the model and simulation sections and clarified the description. Agents in the system are given a random solution in the NK solution space. Starting in round 1, every agent looks at the current states of knowledge (i.e., solutions) of its neighbors (i.e., the other agents it is connected to in the network). Agents then engage in one of two behaviors: exploitation by copying, and, if that fails to produce a better result, exploration. First, by exploitation, the agent will copy the current state of knowledge of a more successful neighbor (i.e., exploits the neighbor's state of knowledge at time t), if there is a more successful neighbor. If none of the agent's neighbors' states of knowledge are currently better at time t , the agent will attempt to explore for better states of knowledge. The agent does so by incrementally changing its current state of knowledge. If its new (read: experimental) state of knowledge that results from a random incremental change has a higher score than its current state of knowledge, the agent adopts this new state of knowledge as its own. Otherwise, the agent remains with what it originally had.

All agents in the network engage in this behavior in a single round. The next round of the simulation ($t+1$) repeats the aforementioned steps, but now agents begin with their current state of knowledge from the previous round t , where they found a new solution from exploring the problem space, copied a neighbor with a better solution, or continued with their status quo solution. This process continues for a number of rounds until the entire system reaches an equilibrium,

which is the point at which no more unique knowledge (i.e., solutions) is being generated by agents in the networks.

To the specific comment of how our previous simulation was set up, our original criterion is that each simulation in the diversity of ability used the same 100 unique starting points. However, we have now changed this, and allow for 100 unique starting points to be used in each simulation, such that no two simulations use the same starting points. However, for each simulation, these starting points are the same across the five intermixing setups. In other words, each intermixing setup for a single simulation uses the same starting points to ensure that every condition is the same except how different species of agents are distributed in the network (i.e., intermixing).

3). There are two species of agent described in the methods section with two non-overlapping methods for generating new solutions. Are these agents that differ in ability? What are agents that differ in knowledge?

Yes, these agents are agents that differ in ability: species A and species B. In our new setup, species A and species B are assigned distinct, but non-overlapping search heuristics for each simulation, instead of having the same search heuristic across every simulation. Now, our new setup is such that species A's search heuristic is different in each simulation (as is species B's), but is always mutually exclusive from B so that they do not overlap.

For simulations that explore the diversity in knowledge, we use the A/B species agents again. However, since calling an agent "A" or "B" is merely a label (i.e., there is nothing inherent about being an "A" agent or a "B" agent), we randomly assign the same search heuristics used by these agents for the diversity of ability simulations to the agents used in the diversity of knowledge simulations. However, all of the agents in the diversity of knowledge simulation have the same search heuristic, instead of 50 agents with one search heuristic and 50 with another as was the case in the diversity of ability simulations. Instead, these diversity of knowledge simulations differ not in the type or distribution of agents, but instead in how their initial solutions are distributed. We have clarified the language in the manuscript to this effect.

4). The specific problems agents solve are not specified in the manuscript. Solutions are strings of "1s" and "0s" of length 20, but how are they introduced? How do agents evaluate the possible solutions they generate?

We have clarified what and how the NK space works to better address the specific problem agents are tasked to solve. (We also explain how we tested the robustness of our results using an alternative problem space called the Travelling Salesperson Problem [TSP] in the Supplemental Information section.) This is explained in greater detail in both the main text and in the Supplemental Information section, where the latter explains more of the mechanics of how agents manipulate the string of 1s and 0s.

How solutions are introduced and evaluated are explained in the methods section in detail, but we clarify the general design principles we used in the main text. Namely, the string of 1s and 0s is the initial solution given to agents at the start of every simulation and, subsequently, the solution that agents manipulate and copy from other agents during the simulation. This is specific to the NK space simulations. In the diversity of ability simulations, each agent is given a unique initial solution.

5). How do other strategies for generating possible solutions do? In the supplementary material, an A+ species is discussed, which flips bits in a string. Why is it only discussed in the supplementary material? If it produces the union of strategies A and B, shouldn't homogenous groups of A+ species do as well as diverse groups of A and B?

Our focus is on the performance of the distribution of diversity, where we introduce two types, local diversity of ability and diversity of knowledge. We always hold equal the system level diversity-- our question, for example, is not whether a group of A+ species would outperform a group of As (it would); rather, we ask whether for a given mixture of As and A+'s is it better for the two species to be interspersed (answer: yes). We reference this result in the main manuscript but need to relegate it to the Supplementary Information section due to space constraints.

6). What computer language and/or agent-based simulation environment was used?

We wrote our code in Python (and made mention of this in our manuscript), and used R to plot our figures. We will publish our code on Dataverse upon publication of this paper.

Reviewer 3

1). The model focuses on a toy situation where you have just A and B. But in most real-life situation there are A, B, C, and D types of diversity. We have the Democrats and Republicans but Independents play an important role too.

We have extended the model to include a wider array of diversity of ability simulation results with five species (instead of two species) in our Supplementary Materials section. We chose five species to show what would happen with more than two diverse species of agents (i.e., our main results) and to mirror our diversity of knowledge setup with five initial

states of knowledge. Furthermore, we do not include a larger number of species than five, as our focus is the distribution of diversity rather than the marginal effect of more diversity. Our overall claim still holds.

We do note, of course, that the world is vastly more complex still; but we do think this is within the spirit of modeling more generally, i.e., to create a model that captures some constrained but essential truth about the world. E.g., Schelling's influential segregation model assumes that there are only two types of people; and that there is a threshold point where an individual decides to move. This is clearly wrong on many dimensions, yet the model is still quite useful.

2). I found myself incapable of identifying applications. The key reason is the level of abstraction of the paper makes the findings nearly uninterpretable and impossible to confidently to apply to a real-life situation. Because the ABM and manipulations are done without any specification of context, it was possible to think of any example that could be both consistent and inconsistent with your model's findings.

For example, let's say there are farmers and ranchers, two groups that differ in skills. A rancher knows that killing wolf packs means less cattle lost to predation. So, other ranchers copy the practice and so do farmers who want to help the ranchers and live on lands in between grazing land. So, if performance is measured as fewer dead cattle then mixing is trivially good because wolves are exterminated quicker and more completely. But, say change the situation is examined from a slightly different perspective, once all the wolves are exterminated, there is an explosion in the deer and buffalo population, which means cattle have less grazing ground and farmers are having their crops eaten. The explosion of deer also brings ticks, which make the ranchers and farmers sick and unable to work, and the general public's health care bills are rising, leading to even greater rather than less costs. So, by arbitrarily specifying a situation, the model can be shown to fit anything, which makes the model unable explain very much in the end. I suppose the counter argument to my criticism that you can't make everyone happy. Nevertheless, if all that you want to improve is the welfare of the ranchers, then I think the problem is pretty easy and you don't need a model to explain it.

The issue you raise is profound, and endemic in the social sciences. Any research that seeks to study the assertion "A causes B" is typically constrained, both in terms of time scale studied and complexity of interactions with other factors. Randomized drug trials tend to look at mortality and morbidity within certain time scales, and for average treatment effects-- any long term effects, or interactions that make the treatment counterproductive in certain subpopulations will typically be missed. If science is working well (and it often does not), then hopefully a critique of a given finding will emerge in subsequent research (e.g., an RCT with a particular subpopulations).

Here we have made a particular effort to build on an abstraction with an ample prior history in the literature: Agent-based models of problem solving in the NK space. In this model, agents are myopically searching in a solution landscape with many local optima, balancing feedback from the environment (success/failure) and input from peers--

something that has broad empirical relevance to human problem solving. Consider: how do restaurants evaluate strategies to become profitable-- in part evaluating what works, and in part by looking at what other successful restaurants are doing. How do faculty figure out strategies to be productive? Schools figure out how to educate their students? These are all situations of very difficult problem solving, where there are processes of direct experience and social learning, where we might think of exogenous factors that would affect local heterogeneity. However, whether the hypothesized relationships exist in these different systems will likely vary enormously, because there will be contextual factors that may affect the hypothesized relationship (e.g., local heterogeneity may lead to social conflicts and less information sharing in some social systems; or functional needs for cross type communication overwhelm any generative upsides).

So, in response to this comment: we have done two things. First, in the introduction and the "Complex Problems" sections, we have bolstered the discussion of scenarios in the world that qualitatively match our model. Second, we now seek to provide a sense of the contextual nature of these results in the discussion section.

Reviewers' comments:

Reviewer #1 (Remarks to the Author):

I found this revised version far clearer than the original submission and would recommend this for publication. The only v.minor formatting detail would be to define error bars in figure legends.

Reviewer #3 (Remarks to the Author):

My central criticism of the paper was its lack of experimental realism. The model is on a level of abstraction that makes it difficult to find a situation in real-life that fits the characteristics of the model in a meaningful way and by that, characteristics that define exclusion or inclusion within the model's parameters and make predictions of actual outcomes.

Said another way, the model is written in a way that it appears to have no clear boundary conditions, and without boundary conditions on the cases that fit and don't fit the model, it is impossible to have confirm or disconfirming evidence or a clear understanding of how to apply the model. I gave several examples of this problematic in my first round review. I noticed that another reviewer made a similar comment.

The authors did not appear to respond in a serious way to these criticisms or do much in general to revise the paper. Rather, the authors responded by saying that a classic study by Schelling had a similar problem. I believe this argument is meant to say that the existence of a famous model justifies the same approach by the authors in their paper. I was not persuaded by this argument. If we evaluate Schelling's 50 year old model by what we know today about segregation, the Schelling model looks like an oversimplification. Lots of factors that became known after the model were left out of the original model. This means that when Schelling modeled segregation 50 years ago his model resembled the reality of his time more closely, and closely enough, to make generative contribution to the subject. So, the comparison between your model and Schellings model seems irrelevant to the issues of realism in your model. I would also add that you are working in a space where there has been a tremendous amount of empirical and theoretical work in the social sciences, economics, humanities, and increasingly in the physical sciences. So, your model feels particularly theoretically and empirically austere.

The Schelling model nevertheless is what I think you should aspire to in your work. However, whereas you appear to be aspiring to be as good as Schelling greatest weakness (by your own account), I think the paper should aspire to be as good as Schelling's greatest strength. His strength was in showing how the features of his model align with real world dynamics and real data. This has also been the case with all the work his work has generated.

So, I think you have a story to tell and the paper can make a contribution except not in its current state. What I would like to see put into the paper is the following (and perhaps I should have been more precise about this in the first round).

I think the paper needs to show how the ABM model maps on to at least one real world case in a meaningful way. By a meaningful way, I mean that the real-life case should be consequential in its impact. That is the organization of knowledge and skills in line with your predictions had the expected change in the marketplace or thinking about problem. For example, can your model explain a collective action like the Manhattan project, the red ball challenge, the Bay of Pigs vs Cuban missile crisis, or other human generated data from surveys or experiments?

Papers that do the type of simulation supported by data analysis I am referring to above include: Dugundji & Gulyás (2008) "Sociodynamic discrete choice on networks in space: impacts of agent heterogeneity on emergent outcomes." *Environment and Planning B: Planning and Design* 35,

1028-1054)

Kaufmann, Stagl, & Franks (2009). "Simulating the diffusion of organic farming practices in two New EU Member States." *Ecological Economics* 2580-2593 (2009)

Hauser, O. P., Rand, D. G., Peysakhovich, A. & Nowak, M. A. Cooperating with the future. *Nature* 511, 220 (2014).

Once you have found a consequential case, provide data that evidences the links between the variables in your model and the variables in the case.

Show that predictions of your model are consistent with the outcomes of the case or other data consistent with the models predictions.

I realize that this approach could be criticized on the grounds that finding one case that is consistent with your model suffers from sampling bias. Nevertheless, one good proof of concept case is better than having no real life instantiation of your model at all. And, if you provide a case, the readers can judge for themselves whether the case you explore suffers from excessive sampling bias.

Reviewer 1

I found this revised version far clearer than the original submission and would recommend this for publication. The only v.minor formatting detail would be to define error bars in figure legends.

We thank Reviewer 1 for their helpful feedback. Following from their comment, we have added in and defined the error bars in the discussion.

“As such, we also include the standard errors from these spaces as error bars in this figure and all subsequent figures.”

Reviewer 3

My central criticism of the paper was its lack of experimental realism. The model is on a level of abstraction that makes it difficult to find a situation in real-life that fits the characteristics of the model in a meaningful way and by that, characteristics that define exclusion or inclusion within the model's parameters and make predictions of actual outcomes.

Said another way, the model is written in a way that it appears to have no clear boundary conditions, and without boundary conditions on the cases that fit and don't fit the model, it is impossible to have confirm or disconfirming evidence or a clear understanding of how to apply the model. I gave several examples of this problematic in my first round review. I noticed that another reviewer made a similar comment.

I think the paper needs to show how the ABM model maps on to at least one real world case in a meaningful way. By a meaningful way, I mean that the real-life case should be consequential in its impact. That is the organization of knowledge and skills in line with your predictions had the expected change in the marketplace or thinking about problem.

For example, can your model explain a collective action like the Manhattan project, the red ball challenge, the Bay of Pigs vs Cuban missile crisis, or other human generated data from surveys or experiments?

Papers that do the type of simulation supported by data analysis I am referring to above include:

Dugundji & Gulyás (2008) “Sociodynamic discrete choice on networks in space: impacts of agent heterogeneity on emergent outcomes.” *Environment and Planning B: Planning and Design* 35, 1028-1054)

Kaufmann, Stagl, & Franks (2009). “Simulating the diffusion of organic farming practices in two New EU Member States.” *Ecological Economics* 2580-2593 (2009)

Hauser, O. P., Rand, D. G., Peysakhovich, A. & Nowak, M. A. Cooperating with the future. *Nature* 511, 220 (2014).

Once you have found a consequential case, provide data that evidences the links between the variables in your model and the variables in the case.

Show that predictions of your model are consistent with the outcomes of the case or other data consistent with the models predictions.

I realize that this approach could be criticized on the grounds that finding one case that is consistent with your model suffers from sampling bias. Nevertheless, one good proof of concept case is better than having no real life instantiation of your model at all. And, if you provide a case, the readers can judge for themselves whether the case you explore suffers from excessive sampling bias.

We agree with Reviewer 3 that we can and should tighten the discussion of boundary conditions, which we now include in the discussion section. In particular, we first added more context to our examples that originally used high school principals. In our update, and per Reviewer 3's helpful suggestion, we found an empirical study of high school superintendents (Torenvlied, R., Akkerman, A., Meier, K. J. & O'Toole, L. J. *The Multiple Dimensions of Managerial Networking*. *Am. Rev. Public Adm.* 43, 251–272 (2013)) and highlighted connections with our model:

“To expound on this a bit further with respect to the NK space, consider superintendents of high school districts. Here, superintendents can learn from the best practices and performance of superintendents from other high school districts, specifically where they can see the practices and performance of other local schools. The improvement or decline of one school or district does not necessarily affect another. Furthermore, superintendents are diverse in that they employ approaches that differ from other superintendents as to how they view the schools in their districts: Some may view their schools more as administrators, while others view it more pedagogically. As such, one “activity” that a superintendent as an administrator might employ at their schools is to hire better teachers or to change to block scheduling. Alternatively, a pedagogically oriented superintendent may work to improve the curriculum collaboratively with local teachers. However, these activities do not exist independently. Instead, they have synergies among them such that this approach only works in the presence of other tactics (e.g., certain curriculum improvements only work with block scheduling.). As such, the NK space captures the interaction of activities and these synergies. In other words, it is contingent on “sub-problems” related to the wider goal.”

In addition, we also added the following paragraph in the discussion to further highlight these boundary conditions:

“These findings come with particular scope conditions, that derive from the model’s assumptions. They are relevant to optimization problems that individuals (imperfectly) solve, typically getting stuck in local optima, but where those individuals can learn from others. Individuals are myopic—i.e., able to produce solutions within a small neighborhood around the current solution. All individuals are attempting to solve the identical problem; but individuals vary in their position in the network, and in their ability to produce new solutions given their knowledge. This potentially encompasses a broad set of human problem solving. Examples of social learning that approximately fit into this paradigm might include superintendents of high school districts (as discussed above); graduate students trying to succeed in their program; or small-business franchises. In each case, the relevant agents are trying to solve a complex problem, and can learn from the behavior of other actors, with whom they are generally not in direct competition with. However, our approach excludes a broad set of collective problem behavior; e.g., where skill in groups is coupled with specialization of roles. These results have little implication, for example, for the success of a baseball team; or for scenarios where agents are solving substantially different problems (e.g., career success in very different fields).”

We also believe, however, that it is problematic for us to reconfigure our paper around a particular empirical application, as suggested by the reviewer. While we note that there is indeed an emergent statistically flavored ABM literature (an excellent development), there is also a longer standing theory-building tradition, very much in the Thomas Schelling tradition. Indeed, most of the prominent ABM pieces today remain in this Schelling tradition. This includes the highly cited Centola and Macy (2007) and Lazer and Friedman (2007) papers, as well the works of Scott Page, one of the thought leaders in the application of diversity to collective problem solving. The criterion of empirical fit that Reviewer 3 offers would exclude all of these papers, and, indeed, most of the contemporary ABM-oriented literature.

A paper just recently published in Nature Communications by Barkoczi and Galesic (2018) is in quite similar territory as our paper conceptually and epistemologically. Basically, it induces a set of theoretical propositions from manipulating parameters in an ABM. Substantively--similar to our paper--it is

about the impact of network structure on the balance between exploration and exploitation, and thus on group performance.

We reached out to the editor for guidance on how to proceed with your suggestion, specifically “how the ABM model maps on to at least one real world case [that] should be consequential in its impact” as you have outlined it here. While we appreciate the thoughtfulness of Reviewer 3’s comments, the editor has indicated that we should stay within the framework of theoretical (as compared to statistical) ABMs. As noted, we have, however, bolstered the discussion of the boundary conditions of the propositions produced by the model.